# De-Escalation of Axillary Surgery in Clinically Node-Positive Breast Cancer Patients Treated with Neoadjuvant Therapy: Comparative Long-Term Outcomes of Sentinel Lymph Node Biopsy versus Axillary Lymph Node Dissection

**DOI:** 10.3390/cancers16183168

**Published:** 2024-09-15

**Authors:** Corrado Tinterri, Erika Barbieri, Andrea Sagona, Simone Di Maria Grimaldi, Damiano Gentile

**Affiliations:** 1Breast Unit, IRCCS Humanitas Research Hospital, Via Manzoni 56, Rozzano, 20089 Milan, Italy; corrado.tinterri@hunimed.eu (C.T.); erika.barbieri@cancercenter.humanitas.it (E.B.); andrea.sagona@cancercenter.humanitas.it (A.S.); simone.dimariagrimaldi@cancercenter.humanitas.it (S.D.M.G.); 2Department of Biomedical Sciences, Humanitas University, Via Rita Levi Montalcini 4, Pieve Emanuele, 20090 Milan, Italy

**Keywords:** breast cancer, neoadjuvant therapy, sentinel lymph node biopsy, axillary lymph node dissection, de-escalation

## Abstract

**Simple Summary:**

This study evaluates the long-term oncological outcomes of sentinel lymph node biopsy (SLNB) compared to axillary lymph node dissection (ALND) in clinically node-positive (cN+) breast cancer patients treated with neoadjuvant therapy (NAT). Analyzing data from 322 patients, the study found that SLNB is associated with better long-term outcomes, including recurrence-free survival (RFS), distant disease-free survival (DDFS), overall survival (OS), and breast cancer-specific survival (BCSS), compared to ALND. The axillary recurrence rates for SLNB ranged between 0.9% and 2.1%, comparable to those reported in other studies. Despite the high false-negative rates previously associated with SLNB, this study’s findings support its safety and effectiveness in this patient population. Overall, SLNB is a viable alternative to ALND, offering better long-term oncological outcomes without increasing the risk of axillary recurrence.

**Abstract:**

Backgrounds: This study compares the long-term outcomes of axillary lymph node dissection (ALND) versus sentinel lymph node biopsy (SLNB) in clinically node-positive (cN+) breast cancer (BC) patients treated with neoadjuvant therapy (NAT).Methods: We conducted a retrospective analysis of 322 cN+ BC patients who became clinically node-negative (ycN0) post-NAT. Patients were categorized based on the final type of axillary surgery performed: ALND or SLNB. Recurrence-free survival (RFS), distant disease-free survival (DDFS), overall survival (OS), and breast cancer-specific survival (BCSS) were evaluated and compared between the two groups. Results: Patients in the SLNB group had significantly better 3-, 5-, and 10-year RFS, DDFS, OS, and BCSS compared to those in the ALND group. The SLNB group also had a higher proportion of patients achieving pathologic complete response (pCR). Multivariate analysis identified pCR, ypN0 status, and SLNB as favorable prognostic factors for all survival metrics. Axillary recurrence rates were low for both groups (0.6–2.1%). Conclusions: SLNB may be a safe and effective alternative to ALND for selected cN+ BC patients who convert to ycN0 after NAT. These findings suggest that careful patient selection is crucial, and further research is needed to validate these results in more comparable populations.

## 1. Introduction

Sentinel lymph node biopsy (SLNB) has become the preferred method, supplanting axillary lymph node dissection (ALND) in breast cancer (BC) surgery, especially for patients with a clinically node-negative (cN0) axilla [1,2]. This less invasive technique offers equivalent staging information as ALND but with significantly reduced arm morbidity [3]. Neoadjuvant therapy (NAT) has become an integral component in the multimodal treatment of BC, offering several clinical benefits, including the facilitation of breast conservation, the ability to evaluate in vivo response, and the potential to reduce axillary disease [4,5]. Just as downstaging the primary breast tumor enhances the likelihood of breast conservation, axillary downstaging in patients with clinically node-positive (cN+) disease may similarly increase the chances of preserving axillary nodes. In cases of advanced loco-regional disease, NAT not only addresses the primary tumor but also aims to reduce the extent of axillary surgery and minimizethe risks associated with ALND [6,7,8,9]. Nonetheless, the role of SLNB and its oncological outcomes in cN+ patients before NAT continues to be debated [10,11,12,13]. The controversy revolves around the appropriateness of performing SLNB in this group, as pre-operative systemic treatment may affect the lymphatic pathways of the breast [14,15,16]. Currently, SLNB is employed for axillary staging in BC patients who are cN0 prior to NAT, with studies showing false-negative rates (FNR) of approximately 10% and acceptable sentinel lymph node (SLN) detection rates [17,18,19,20]. The inclusion of targeted therapies such as Trastuzumab, Pertuzumab, and Pembrolizumab, especially in human epidermal growth factor receptor 2 (HER2)-positive or triple-negative BC, along with anthracycline and taxane-based chemotherapy regimens, has demonstrated significant potential in eradicating nodal involvement in approximately 50–75% of cases, depending on the biological subtype of BC [21,22,23,24,25]. Despite these advancements, the implementation of SLNB after NAT for patients initially diagnosed with cN+ disease has been called into question, as earlier studies have reported FNRs ranging from 8.4% to 23.9% [7,8,26,27,28,29]. In response to these concerns, target axillary dissection (TAD) has been introduced as a method to lower the FNR. This approach includes marking positive axillary nodes before NAT with a tattoo or metallic clip, employing a dual tracer, or removing a minimum of three SLNs [30,31,32]. Despite these efforts, studies have indicated that such interventions do not necessarily correlate with prognostic improvements, as the use of SLNB has not been associated with increased rates of axillary recurrences or poorer oncological outcomes [33,34,35,36,37]. In our study, we aimed to evaluate the characteristics and long-term outcomes of axillary node-positive BC patients treated with NAT. This study specifically focuses on comparing the oncological outcomes between two distinct surgical approaches: ALND versus SLNB. Additionally, we sought to identify prognostic factors, including axillary nodal status post-NAT (ypN0 versus ypN+), that may influence recurrence and survival in this patient population.

## 2. Materials and Methods

### 2.1. Patient Selection and Pre-Operative Assessment

We performed a retrospective analysis of all consecutive axillary node-positive BC patients scheduled for NAT from January 2009 to December 2021 at the Breast Unit of IRCCS Humanitas Research Hospital in Milan, Italy. Comprehensive data collection included demographic data, baseline, and post-NAT data, namely age and menopausal status, together with detailed baseline and post-NAT clinical tumor characteristics, including stage, size, focality, subtype, histotype, and the presence of vascular invasion. In the current study, all BC patients were pre-operatively staged for breast and axillary tumor extension by clinical examination in association with imaging studies that included bilateral breast and axillary ultrasound (US). The pre-operative diagnosis of invasive BC was established through US-guided core needle biopsy. Although bilateral mammography, magnetic resonance imaging (MRI) of the breast, and positron emission tomography (PET) scans were not required, they were conducted in most cases. Additionally, the majority of patients underwent US-guided core needle biopsy of the clinic–radiological suspected metastatic axillary lymph nodes before NAT. In this study, cN+ BC patients were defined as those with a positive core needle biopsy and/or palpable axillary lymph nodes prior to NAT and positive findings on axillary US, MRI, or PET scans. The treatment plan for each cN+ BC patient scheduled for NAT was reviewed by a multidisciplinary tumor board, including breast surgeons, oncologists, pathologists, radiotherapists, radiologists, geneticists, and plastic surgeons. Before surgery, clinical and radiological re-staging after NAT was essential to monitor the response to systemic treatment and assess residual tumor size. Physical examinations were performed at the completion of each cycle of chemotherapy, and all patients were thoroughly re-evaluated with a bilateral breast and axillary US after three months of NAT. Those who had previously been evaluated with a PET or MRI repeated the same imaging modality to have uniformity of their follow-up assessment.

### 2.2. Surgical Procedures and Exclusion Criteria

The final NAT cycle was administered strictly within the preceding 30 days of surgery. All patients were found to have a ycN0 status after NAT and subsequently received surgical intervention. Patients received either breast conservation surgery or mastectomy for breast surgical management. Regarding axillary surgical treatment, patients were either subjected to ALND or SLNB. Between late 2017 and early 2018, a transition occurred where single tracer SLNB became the standard method for axillary staging in cN+ BC patients treated with NAT who converted to ycN0. Before this change, direct ALND was the predominant approach, though in some cases, SLNB was already being employed at the discretion of the breast surgeon. The decision to perform SLNB or ALND was not influenced by tumor subtype. For those scheduled for SLNB, lymphatic mapping was conducted using only a single tracer radioisotope, and TAD was not employed. For cases with palpable tumors, our standard protocol involved the peri-tumoral administration of ^99^Technetium-labeled radiocolloid, which was used to trace the intra-mammary lymphatic channels that may be involved in metastatic spread. For non-palpable or multicentric tumors, the radiocolloid was injected dermally into the sub-areolar plexus. Lymphoscintigraphy was carried out pre-operatively, either the day before or on the day of surgery for mapping the lymphatic drainage. A gamma probe was inserted through the axillary incision and guided the dissection until the SLN was identified. In all patients treated with SLNB, at least one SLN was successfully identified. Upon identification with the gammaprobe, the SLN was excised and subjected to a complete intra-operative frozen section pathological examination. The SLN specimens were promptly delivered to the Department of Pathology for frozen section analysis. The adipose tissue surrounding the lymph nodes was carefully removed, and the lymph nodes were sliced along their long axis. At least three levels of the frozen sections were examined by a breast-specialized pathologist with the assistance of a trainee pathologist. The remaining unfrozen portions of the lymph nodes were subsequently fixed in formalin, embedded in paraffin, and subjected to further histopathological examination to confirm the initial findings. In patients with a ypN0 status following SLNB, further axillary clearance was omitted. Complete axillary dissection was performed only if the SLN was found to contain residual disease during the intra-operative pathological assessment. Pathologic complete response (pCR) was defined as not invasive or non-invasive residual tumor in both breast and axillary nodes (ypT0 N0), excluding patients with pathological stage ypTis. The exclusion criteria for this study included BC patients scheduled for up-front surgery, those with clinic–radiological detectable metastases (cM+), cN0 BC patients scheduled for NAT, patients remaining ycN+ post-NAT, those with disease progression during NAT, patients with a history of other malignancies, those with a follow-up period of less than 32 months, and patients who were lost to follow-up. Each patient provided informed consent for surgery and the collection of clinical data.

### 2.3. Statistical Analysis

Patients were selected from our prospectively maintained institutional database, the last follow-up of which was updated until 30 July 2024. The patient characteristics were categorized based on the type of axillary surgery received and presented as the median and range for continuous variables, as well as frequencies (number, percentage) for categorical variables. Differences in demographic, clinicopathologic, and treatment characteristics between the two groups were assessed using the chi-square test. Variables that were significant in the univariate analysis were further evaluated using multivariate logistic regression to rule out confounding variables. For recurrence and survival analyses, patients were categorized based on their final type of axillary surgical treatment (ALND versus SLNB) and their post-NAT axillary nodal status (ypN0 versus ypN+). Recurrence-free survival (RFS) was calculated from the date of surgery to the first instance of tumor progression, whether loco-regional recurrence or distant metastasis. Distant disease-free survival (DDFS) was determined from the surgery date until the initial appearance of distant metastasis. Overall survival (OS) was calculated from the time of surgery to death from any cause or to the date of the most recent follow-up. Breast cancer-specific survival (BCSS) was determined by identifying BC as the cause of death, with follow-up time adjusted for deaths due to other causes. Recurrence and survival curves were also created using the Kaplan–Meier method, from which RFS, DDFS, OS, and BCSS rates were estimated. The log-rank test was conducted to compare recurrence and survival distributions between these groups. Next, a multivariate Cox regression analysis was applied to find the independent risk factors associated with recurrence and survival. All statistical tests were two-sided, and *p*-values less than 0.05 were considered statistically significant. The descriptive statistics analyses and visualizations were performed using IBM SPSS Statistics software, version 25.0. SLNB vs. ALND_dataset can be found in the Appendix A.

## 3. Results

### 3.1. Baseline and Post-Neoadjuvant Therapy Patient Characteristics and Axillary Surgical Interventions

A total of 322 cN+ BC patients who underwent NAT were included in this study. The median age of the patients was 51 years (range 20–87), with 178 patients (55.3%) being post-menopausal. Pre-operative imaging and staging were comprehensive, with mammography performed in 229 patients (71.1%) and US imaging of the breast and axilla conducted in all 322 patients (100%). Axillary biopsy was conducted in 194 patients (60.3%). The median tumor size prior to NAT was 35 mm (range 7–100). The majority of patients (226; 70.2%) presented with a single nodule. The distribution of tumor subtypes revealed that 112 patients (34.8%) had luminal-like tumors, 136 patients (42.2%) had HER2-positive tumors, and 74 patients (23.0%) had triple-negative BC. Vascular invasion was present in 76 patients (23.6%). Following NAT, the median tumor size was reduced to 9 mm (range 0–100), and 84 patients (26.1%) achieved a pCR. For breast surgery, 150 patients (46.6%) underwent breast-conserving surgery, while 172 patients (53.4%) underwent mastectomy. Post-operatively, 183 patients (56.8%) received endocrine therapy, 95 patients (29.5%) were treated with Trastuzumab–emtansine (T-DM1), 25 patients (7.8%) with taxanes, and 19 patients (5.9%) with capecitabine. Demographic, clinic–pathologic characteristics, and treatments received are shown in Table 1.

Regarding intra-operative axillary staging, 159 patients (49.4%) underwent SLNB, while 163 patients (50.6%) underwent direct ALND. Among those who had SLNB, 112 patients (70.4%) did not require subsequent axillary clearance, whereas 47 patients (29.6%) underwent SLNB followed by ALND. In the ALND group, the median number of non-SLNs evaluated was 14 (range 5–49), while in the SLNB group, the median number of SLNs identified was 2 (range 1–6).The median number of positive non-SLNs was one (range 0–34). Axillary lymph nodecharacteristics of patients are detailed in Table 2.

### 3.2. Comparative Analysis of Patient Characteristics between Direct Axillary Lymph Node Dissection and Sentinel Lymph Node Biopsy Groups

The demographic, clinicopathologic, and treatment-related characteristics of the 322 patients were analyzed and compared between the two surgical groups: direct ALND versus SLNB. In the univariate analysis, age distribution did not differ significantly between the direct ALND and SLNB groups (*p* = 0.922). Similarly, menopausal status showed no significant difference between the two groups (*p* = 0.122). Pre-operative tumor size was comparable between the groups (*p* = 0.303), as was the presence of a single nodule (*p* = 0.064). The clinical stage before NAT (cT1-2 versus cT3-4) also did not show a significant difference between the direct ALND and SLNB groups (*p* = 0.971). However, a significant difference was observed in the type of breast surgery performed, with the direct ALND group more likely to undergo mastectomy compared to the SLNB group (69.3% versus 37.1%, respectively; *p* < 0.0001). This finding was further supported by multivariate analysis, where undergoing breast-conserving surgery was significantly associated with the SLNB group [Odds ratio (OR) = 0.255, 95% Confidence interval (95% CI): 0.154–0.421, *p* < 0.0001]. In terms of tumor subtype, a significant difference was noted in the distribution of luminal-like, HER2-positive, and triple-negative subtypes (*p* = 0.014), with the direct ALND group having a higher proportion of luminal-like tumors. Post-NAT tumor size showed a significant difference between the groups, with larger residual tumors more common in the direct ALND group (*p* = 0.010). The axillary stage after NAT also differed significantly, with the SLNB group having a higher rate of ypN0 status (63.5% versus 36.8% in the direct ALND group; *p* < 0.0001). This difference was reflected in the multivariate analysis, where the likelihood of ypN0 status was significantly higher in the SLNB group (OR = 0.546, 95% CI: 0.436–0.684, *p* < 0.0001). Univariate and multivariate analyses are summarized in Table 3.

### 3.3. Long-Term Oncological Outcomes and Prognostic Factors for Recurrence and Survival

At a median follow-up of 75 months (range 32–189), axillary recurrence was documented in three patients across different treatment groups. One patient from the direct ALND group (1/163, 0.6%) had a ypN1a axillary stage and received adjuvant axillary radiotherapy. Another patient who underwent SLNB followed by subsequent axillary clearance (1/47, 2.1%) also had a ypN1a axillary stage but did not receive adjuvant axillary radiotherapy. The third patient, who underwent SLNB without further axillary clearance (1/112, 0.9%), had a ypN0 status and was treated with adjuvant axillary radiotherapy. Among these three cases, two recurrences were synchronous with a recurrence in the skin, occurring in patients who had previously undergone mastectomy. The third recurrence was isolated to the axilla. Management of these recurrences varied, with two patients undergoing a second axillary surgery, while the remaining patient was treated exclusively with systemic therapy. Overall, a total of 58 patients (18.0%) died during the follow-up period. This included 47 patients from the direct ALND group (47/163, 28.8%) and 11 patients from the SLNB group (11/159, 6.9%). To evaluate the long-term oncological outcomes, patients were stratified according to the final type of axillary surgery performed—SLNB versus ALND—and their RFS, DDFS, OS, and BCSS rates were compared. The RFS rates at 3, 5, and 10 years were significantly higher in the SLNB group, recorded at 90.8%, 84.7%, and 84.7%, respectively, compared to 77.1%, 73.2%, and 63.1% in the ALND group (*p* = 0.001). Similarly, DDFS rates at 3, 5, and 10 years were 93.6%, 87.9%, and 84.7% in the SLNB group, compared to 79.0%, 74.5%, and 63.1% in the ALND group (*p* = 0.001). When considering OS, patients in the SLNB group again demonstrated superior outcomes, with 3-, 5-, and 10-year survival rates of 96.3%, 93.3%, and 93.3%, respectively, versus 85.4%, 80.0%, and 72.3% in the ALND group (*p* = 0.001). The BCSS rates followed a similar trend, with the SLNB group achieving 98.1%, 95.0%, and 95.0% survival at 3, 5, and 10 years, respectively, compared to 88.2%, 84.7%, and 79.5% in the ALND group (*p* = 0.002). Statistical analysis revealed that the differences in RFS, DDFS, OS, and BCSS between the SLNB and ALND groups were highly significant. These long-term oncological outcomes are further illustrated in Kaplan–Meier recurrence and survival curves, as shown in Figure 1 and Figure 2.

To assess another significant independent factor potentially impacting long-term oncological outcomes, patients were stratified by their axillary stage post-NAT—ypN0 versus ypN+—and their RFS, DDFS, OS, and BCSS rates were compared. In the ypN0 group, 23 cases of recurrence (23/161, 14.3%) were observed, with 9 recurrences occurring in the ALND group and 14 in the SLNB group. In terms of loco-regional recurrences within the ypN0 cohort, four cases were noted in the ipsilateral breast, one in the ipsilateral axilla, one in the contralateral axilla, and one in the skin. In contrast, within the ypN+ group, 54 recurrences (54/161, 33.5%) were documented, all of which occurred in patients who had previously undergone ALND. The loco-regional recurrences in the ypN+ group included three in the ipsilateral breast, one in the contralateral breast, six in the skin, and two in the axilla. The RFS rates at 3, 5, and 10 years were markedly higher in the ypN0 group, recorded at 88.7%, 85.3%, and 79.2%, respectively, compared to 74.9%, 69.5%, and 59.5% in the ypN+ group (*p* < 0.0001). Similarly, the DDFS rates were 90.6%, 87.3%, and 79.2% in the ypN0 group versus 77.4%, 71.1%, and 59.5% in the ypN+ group (*p* < 0.0001). OS rates also favored the ypN0 group, with 3-, 5-, and 10-year survival rates of 95.6%, 93.2%, and 90.2%, respectively, compared to 82.8%, 75.9%, and 67.0% in the ypN+ group (*p* < 0.0001). The BCSS rates followed this pattern, with the ypN0 group achieving 96.2%, 94.5%, and 94.5% survival at 3, 5, and 10 years, respectively, compared to 87.0%, 81.9%, and 74.7% in the ypN+ group (*p* < 0.0001). These long-term oncological outcomes are further illustrated in Kaplan–Meier recurrence and survival curves, as shown in Figure 3 and Figure 4.

Multivariate analysis revealed several factors significantly associated with RFS, DDFS, OS, and BCSS. For RFS, patients who achieved a pCR after NAT had a significantly lower risk of recurrence compared to those with a partial response [Hazard ratio (HR) = 0.282, 95% CI: 0.097–0.815, *p* = 0.019]. Additionally, patients with a post-NAT axillary stage of ypN0 had a more favorable RFS compared to those with residual nodal disease (HR = 1.281, 95% CI: 1.020–1.611, *p* = 0.034). Direct ALND was also associated with a poorer RFS (HR = 0.356, 95% CI: 0.190–0.667, *p* = 0.001). For DDFS, patients who achieved ypN0 status had significantly better DDFS than those with residual nodal disease (HR = 1.285, 95% CI: 1.022–1.617, *p* = 0.032). As with RFS, direct ALND was associated with poorer DDFS (HR = 0.376, 95% CI: 0.200–0.708, *p* = 0.002). In terms of OS, the achievement of pCR was associated with a significantly lower risk of mortality (HR = 0.041, 95% CI: 0.005–0.351, *p* = 0.004). Moreover, ypN0 status was a favorable prognostic indicator for OS (HR = 1.493, 95% CI: 1.145–1.946, *p* = 0.003). Conversely, vascular invasion significantly increased the risk of mortality (HR = 2.497, 95% CI: 1.145–1.946, *p* = 0.005). Finally, for BCSS, those who were ypN0 post-NAT had significantly improved BCSS compared to those with residual nodal disease (HR = 1.493, 95% CI: 1.086–2.053, *p* = 0.014). Direct ALND remained a strong predictor of poorer BCSS (HR = 0.228, 95% CI: 0.083–0.626, *p* = 0.004). Prognostic factors for recurrence and survival are summarized in Table 4.

## 4. Discussion

The findings of our retrospective analysis add to the ongoing debate regarding the optimal surgical management of the axilla in cN+ BC patients who achieve ycN0 status after NAT. While our results are consistent with existing evidence on the oncologic safety of single tracer SLNB in this context, it is crucial to interpret these findings with caution due to the inherent differences between the patient groups. Specifically, the observed differences in tumor biology and response to NAT between the SLNB and ALND groups may have influenced the outcomes. The significant difference in OS between the SLNB and ALND groups suggests that factors beyond the surgical approach likely contribute to these outcomes. For instance, the higher proportion of patients achieving a pCR in the SLNB group compared to the ALND group could partly explain the better outcomes observed in the SLNB group. However, it is unlikely that this alone accounts for the entire difference, indicating that other independent prognostic factors may also be at play. The axillary recurrence rates in our study, ranging between 0.6% and 2.1%, are comparable to those reported in other studies, which vary from 0% to 2.3% [33,34,35,36,37]. However, while our data suggest that SLNB may be associated with better long-term oncological outcomes, including RFS, DDFS, OS, and BCSS, these findings should not be taken to imply that SLNB is universally superior to ALND. Rather, our results suggest that SLNB could be a reasonable option for carefully selected patients who respond well to NAT and convert to ycN0 status.

The observed difference in the type of breast surgery performed between the ALND and SLNB groups, with a higher rate of mastectomies in the ALND group, can be attributed to differences in post-NAT tumor size. Specifically, the ALND group had a significantly larger residual tumor size post-NAT compared to the SLNB group (*p* = 0.010). This likely influenced the surgical decision-making process, leading to a preference for mastectomy over breast-conserving surgery in these patients. The multivariate analysis further supports this finding, as it demonstrated that patients who underwent SLNB were significantly more likely to have breast-conserving surgery (OR = 0.255, 95% CI: 0.154–0.421, *p* < 0.0001). This suggests that the extent of residual disease post-NAT played a critical role in determining the surgical approach.

Historically, SLNB has not been recommended as a viable alternative to ALND for cN+ BC patients who achieve nodal downstaging to ycN0 following NAT. This hesitancy primarily stems from earlier studies that reported unacceptably high FNRs for SLNB in this particular patient population, often exceeding 10% [7,27,28,29]. One of the most important studies in this matter was the American College of Surgeons Oncology Group Z1071 trial [7], a prospective clinical study that enrolled 756 cN+ patients undergoing NAT across 136 institutions. Of these, 649 patients completed the protocol, which involved both SLNB and subsequent axillary dissection after pre-operative systemic therapy. The trial reported a FNR of 12.6%, a figure that raised concerns about the reliability of SLNB as a sole staging procedure in this setting. The SENTINA trial [29] further attempted to answer this question in a prospective, multicenter design involving 103 institutions and 1737 patients. This trial enrolled patients with cN+ disease for which, after administration of NAT, they converted into ycN0 (arm C). These patients underwent SLNB followed by axillary dissection. The pooled data showed a FNR of 24.3% in patients with one removed SLN, which decreased to 18.5% when two SLNs were removed. Adding to this literature, a systematic review and meta-analysis [27] pooled data from eight studies dealing with the use of SLNB after NAT in cN+ BC patients. The pooled estimate of the FNR was given at 15.1%. It is of note that when only one SLN was removed, there was a significantly higher FNR compared to two or more SLNs (23.9% versus 10.4%, *p* = 0.026), highlighting the importance of SLN count in minimizing false negatives. Another analogous systematic review and meta-analysis [28], which included 20 unique studies comprising a total of 2217 patients, reported an overall FNR of 17%. These consistently high FNRs across multiple studies and meta-analyses have historically contributed to the reluctance to adopt SLNB as a standard procedure for cN+ patients post-NAT, as the risk of undetected residual nodal disease remained a significant concern.

Despite various efforts to reduce the FNR associated with SLNB in cN+ BC patients treated with NAT, it is important to note that none of these studies have thoroughly investigated the long-term oncological outcomes. The primary focus of these efforts has been on procedural accuracy rather than prognostic implications. In contrast, our study has extended the analysis to include the long-term oncological outcomes, demonstrating that SLNB is not only a safe procedure but is also associated with a more favorable prognosis in cN+ BC patients treated with NAT. This conclusion is supported by similar findings from other studies with extended follow-up periods. For instance, in a retrospective study conducted at the European Institute of Oncology, which included 688 consecutive patients with cT1-3, cN0-cN+ disease who became or remained ycN0 after NAT and subsequently underwent SLNB, the axillary failure rate among initially cN+ patients was a mere 1.8%. Moreover, the OS rates for the entire cohort were reported to be 91.3% at 5 years and 81.0% at 10 years [33]. Likewise, a recent retrospective study of 610 cT1-3, cN+ patients who achieved ycN0 status following NAT and underwent SLNB further confirmed the safety and effectiveness of the procedure. With a median follow-up of 40 months, the study recorded only one instance of nodal recurrence, which occurred simultaneously with a local recurrence in a patient who had declined adjuvant radiotherapy. The DDFS and OS rates at 5 years were 92.7% and 94.2%, respectively [34]. Moreover, a retrospective study by Piltin et al. [36] evaluated 602 cN+ BC patients treated with NAT and observed a significant increase in the use of SLNB, rising from 21.1% in 2009–2012 to 75.3% in 2015–2019 (*p* < 0.001). Among these patients, 50.5% avoided dissection, with a low regional recurrence rate of 0.9% after SLNB. The study by Keelan et al. [38] systematically reviewed and meta-analyzed data from seven retrospective cohort studies, encompassing a total of 915 patients with histologically confirmed cN+ BC treated with NAT. The pooled analysis revealed a 5-year RFS of 86.5% (95% CI: 82.15–90.35) and a 5-year OS of 93.1% (95% CI: 87.8–97.03) for patients who achieved a negative SLNB after NAT and did not undergo ALND. The recent retrospective study by Lim et al. [39] evaluated 902 BC patients with initially cN+ disease who underwent NAT. Of these, 477 (52.9%) achieved a complete pathological response in the axilla. The study found that among patients who underwent SLNB, the axillary recurrence rate was 3.2%, compared to 1.8% in those who underwent ALND (*p* = 0.398). Notably, RFS and OS were significantly better in the SLNB group, with HR of 0.501 (95% CI: 0.293–0.856, *p* = 0.011) and 0.244 (95% CI: 0.061–0.979, *p* = 0.047), respectively, suggesting that ALND can be safely omitted in this patient population without compromising long-term oncological outcomes.

It should be underlined that our study has some limitations. One of the main limitations of this study is that not all cN+ patients were able to undergo US-guided axillary biopsy, which may have led to an incomplete assessment of nodal disease at baseline. Additionally, the retrospective nature of the study inherently subjects it to selection bias and limits the ability to establish causality. The study was conducted at a single institution, which may restrict the generalizability of the findings to other clinical settings with differing practices and patient populations. Despite its limitations, this study has notable strengths, particularly the long median follow-up period of 75 months, which provides a thorough evaluation of long-term oncological outcomes, and the comprehensive analysis of prognostic factors, which provide a solid foundation for the study’s results regarding the safety and efficacy of SLNB in this patient population.

## 5. Conclusions

In conclusion, this study suggests that SLNB may be a safe and effective alternative to ALND for selected cN+ BC patients who become ycN0 after NAT. While the findings indicate that SLNB could be associated with better long-term oncological outcomes without increasing the risk of axillary recurrence, these results should be applied with caution in clinical practice. The differences between the SLNB and ALND patient groups highlight the need for careful patient selection and further research to validate these findings in more comparable populations.

## Figures and Tables

**Figure 1 cancers-16-03168-f001:**
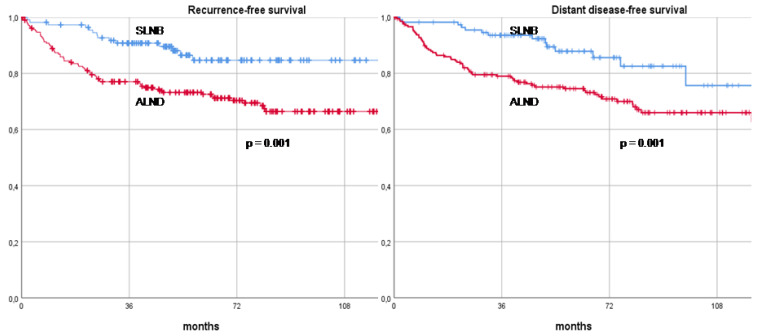
Curves depicting recurrence-free survival and distant disease-free survival in clinically node-positive breast cancer patients who underwent neoadjuvant therapy and axillary surgery (sentinel lymph node biopsy versus axillary lymph node dissection). Footnotes: SLNB: Sentinel lymph node biopsy, ALND: Axillary lymph node dissection.

**Figure 2 cancers-16-03168-f002:**
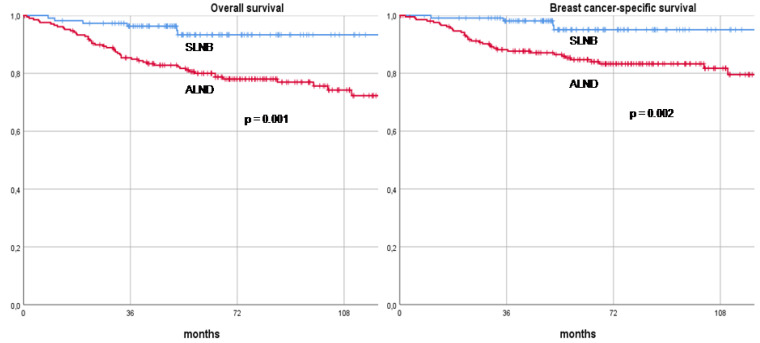
Overall survival and breast cancer-specific survival curves of clinically node-positive patients with breast cancer treated with neoadjuvant therapy and axillary surgery (sentinel lymph node biopsy versus axillary lymph node dissection). Footnotes: SLNB: Sentinel lymph node biopsy, ALND: Axillary lymph node dissection.

**Figure 3 cancers-16-03168-f003:**
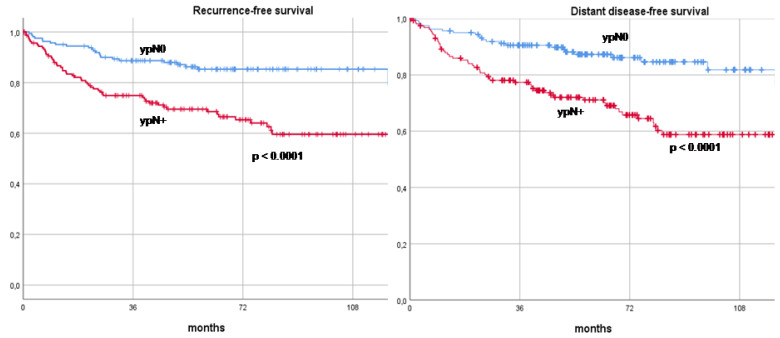
Comparison of recurrence-free survival and distant disease-free survival curves between ypN0 and ypN+ patients. Footnotes: ypN0: Complete pathologic axillary response, ypN+: Residual axillary disease.

**Figure 4 cancers-16-03168-f004:**
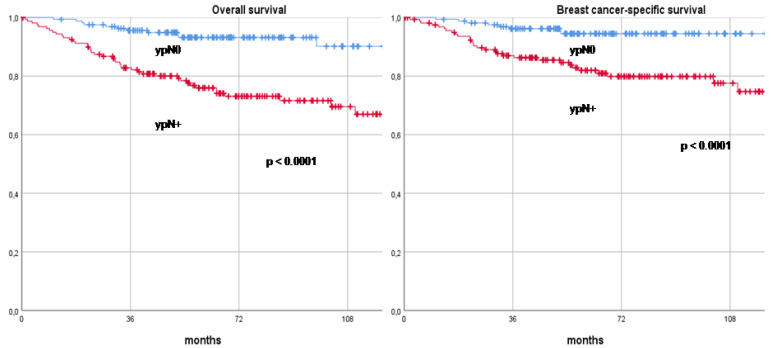
Comparison of overall survival and breast cancer-specific survival curves between ypN0 and ypN+ patients. Footnotes: ypN0: Complete pathologic axillary response, ypN+: Residual axillary disease.

**Table 1 cancers-16-03168-t001:** Baseline and post-neoadjuvant therapy characteristics of clinically node-positive breast cancer patients.

Characteristics	Number (%)/Median (Range)
Patients Age (years) Post-menopausal Pre-operative staging Mammography Breast and axillary US Axillary biopsy MRI PET Size pre-NAT (mm) Single nodule Stage pre-NAT - cT1 - cT2 - cT3 - cT4 - cN+ - NAT with anthracycline only - NAT with anthracycline and taxanes - Trastuzumab - Pertuzumab Tumor Subtype - Luminal-like - HER2-positive - Triple-negative Histotype - Ductal - Lobular - Mucinous - Papillary - Apocrine Vascular invasion pCR Size post-NAT (mm) Stage post-NAT - ypT0 - ypTis - ypTmi - ypT1a - ypT1b - ypT1c - ypT2 - ypT3 - ypT4 - ypN0 - ypNitc/mi - ypN1 - ypN2 - ypN3 Breast surgery - BCS - Mastectomy Post-operative treatment - Taxanes - Capecitabine - Breast radiotherapy - Chest wall radiotherapy - Axillary radiotherapy - Endocrine - T-DM1	51 (20–87) 178 (55.3%) 229 (71.1%) 322 (100%) 194 (60.3%) 133 (41.3%) 236 (73.3%) 35 (7–100) 226 (70.2%) 47 (14.6%) 198 (61.5%) 48 (14.9%) 29 (9.0%) 322 (100%) 35 (10.9%) 272 (84.5%) 133 (41.3%) 2 (0.6%) 112 (34.8%) 136 (42.2%) 74 (23.0%) 296 (91.9%) 18 (5.6%) 5 (1.6%) 2 (0.6%) 1 (0.3%) 76 (23.6%) 74 (23.0%) 9 (0–100) 84 (26.1%) 26 (8.1%) 9 (2.8%) 20 (6.2%) 38 (11.8%) 66 (20.5%) 55 (17.1%) 17 (5.3%) 7 (2.1%) 161 (50.0%) 15 (4.7%) 70 (21.7%) 45 (14.0%) 31 (9.6%) 150 (46.6%) 172 (53.4%) 25 (7.8%) 19 (5.9%) 150 (46.6%) 26 (8.1%) 238 (73.9%) 183 (56.8%) 95 (29.5%)

Footnotes: US: Ultrasound, MRI: Magnetic resonance imaging, PET: Positron emission tomography, NAT: Neoadjuvant therapy, HER2: HER2 evaluated either on immunohistochemistry or on in situ hybridization, according to the ASCO CAP guidelines, pCR: Pathologic complete response, BCS: Breast-conserving surgery, T-DM1: Trastuzumab–emtansine.

**Table 2 cancers-16-03168-t002:** Intra-operative axillary staging in clinically node-positive breast cancer patients treated with neoadjuvant therapy.

	Number (%)/Median (Range)
Intra-operative axillary staging SLNB Direct ALND SLNB not followed by ALND SLNB followed by ALND Axillary data Number of SLNs Number of evaluated non-SLNs Number of positive non-SLNs	159 (49.4%) 163 (50.6%) 112/159 (70.4%) 47/159 (29.6%) 2 (1–6) 14 (5–49) 1 (0–34)

Footnotes: SLNB: Sentinel lymph node biopsy, ALND: Axillary lymph node dissection, SLN: Sentinel lymph node.

**Table 3 cancers-16-03168-t003:** Comparison of patient characteristics between direct axillary lymph node dissection and sentinel lymph node biopsy groups.

Characteristics	dALND (No. 163) Tot. (%)	SLNB (No. 159) Tot. (%)	Univariate Analysis *p*-Value	Multivariate Analysis *p*-Value OR (95% CI)
Demographic Age (years) - ≤51 - >51 Menopausal status - Pre-menopausal - Post-menopausal Pre-operative staging Dimension pre-NAT (mm) - ≤35 - >35 Single nodule - Yes - No Stage pre-NAT - cT1-2 - cT3-4 Surgery - BCS - Mastectomy Tumor Histotype - Ductal - Lobular - Mucinous - Papillary - Apocrine Subtype - Luminal-like - HER2-positive - Triple-negative Dimension post-NAT (mm) - ≤9 - >9 Axillary stage post-NAT - ypN0 - ypNitc/mi - ypN1 - ypN2 - ypN3 pCR - Yes - No Vascular invasion - Yes - No	587 (53.4%) 76 (46.6%) 66 (40.5%) 97 (59.5%) 105 (64.4%) 58 (35.6%) 122 (74.8%) 41 (25.2%) 123 (75.9%) 39 (24.1%) 50 (30.7%) 113 (69.3%) 146 (89.6%) 11 (6.7%) 4 (2.5%) 1 (0.6%) 1 (0.6%) 69 (42.3%) 59 (36.2%) 35 (21.5%) 76 (46.6%) 87 (53.4%) 60 (36.8%) 2 (1.2%) 41 (25.2%) 32 (19.6%) 28 (17.2%) 40 (24.5%) 123 (75.5%) 51 (31.3%) 112 (68.7%)	84 (52.8%) 75 (47.2%) 78 (49.1%) 81 (50.9%) 111 (69.8%) 48 (30.2%) 104 (65.4%) 55 (34.6%) 121 (76.1%) 38 (23.9%) 100 (62.9%) 59 (37.1%) 150 (94.3%) 7 (4.4%) 1 (0.6%) 1 (0.6%) 0 (0%) 43 (27.0%) 77 (48.4%) 39 (24.5%) 97 (61.0%) 62 (39.0%) 101 (63.5%) 13 (8.2%) 29 (18.2%) 13 (8.2%) 3 (1.9%) 34 (21.4%) 125 (78.6%) 25 (15.7%) 134 (84.3%)	0.922 - 0.122 - 0.303 - 0.064 - 0.971 - <0.0001 ^a^ - 0.449 - - - - 0.014 ^a^ - - 0.010 ^a^ - <0.0001 ^a^ - - - - 0.501 0.001 ^a^ -	- - - - - <0.00001 ^a^ 0.255 (0.154–0.421) - - 0.924 1.017 (0.719–1.439) - - 0.090 1.684 (0.922–3.077) - <0.0001^a^ 0.546 (0.436–0.684) - - - - - 0.609 0.837 (0.424–1.654) -

Footnotes: dALND: Direct axillary lymph node dissection, SLNB: Sentinel lymph node biopsy, OR: Odds ratio, 95% CI: 95% Confidence interval, NAT: Neoadjuvant therapy, BCS: Breast-conserving surgery, HER2: HER2 evaluated either on immunohistochemistry or on in situ hybridization, according to the ASCO CAP guidelines, pCR: Pathologic complete response, ^a^: Statistically significant.

**Table 4 cancers-16-03168-t004:** Independent prognostic factors for recurrence and survival in clinically node-positive breast cancer patients treated with neoadjuvant therapy.

Independent Factors	RFS HR (95%CI) *p*-Value	DDFS HR (95%CI) *p*-Value	OS HR (95%CI) *p*-Value	BCSS HR (95%CI) *p*-Value
Patient				
Age (years)				
- ≤51	Reference	Reference	Reference	Reference
- >51	0.894 (0.432–1.848) 0.762	0.942 (0.452–1.964) 0.873	1.593 (0.684–3.771) 0.281	0.829 (0.314–2.187) 0.705
Menopausal status				
- Pre-menopausal	Reference	Reference	Reference	Reference
- Post-menopausal	0.899 (0.442–1.830) 0.769	0.859 (0.418–1.766) 0.680	0.835 (0.354–1.973) 0.681	1.107 (0.413–2.967) 0.840
Pre-operative staging				
Dimension pre-NAT (mm)				
- ≤35	Reference	Reference	Reference	Reference
- >35	1.401 (0.851–2.307) 0.185	1.345 (0.812–2.227) 0.250	0.696 (0.361–1.342) 0.280	0.735 (0.346–1.559) 0.422
Single nodule				
- Yes	Reference	Reference	Reference	Reference
- No	1.213 (0.703–2.091) 0.488	1.268 (0.732–2.197) 0.396	1.341 (0.708–2.540) 0.368	1.006 (0.456–2.219) 0.989
Stage pre-NAT				
- 2	Reference	Reference	Reference	Reference
- 4	0.903 (0.509–1.603) 0.727	0.918 (0.517–1.631) 0.771	0.951 (0.476–1.902) 0.888	1.561 (0.707–3.445) 0.270
Tumor				
Histotype				
- Ductal	Reference	Reference	Reference	Reference
- Other	1.056 (0.608–1.835) 0.847	0.983 (0.570–1.693) 0.950	0.780 (0.384–1.586) 0.493	1.108 (0.500–2.455) 0.801
Subtype				
- Luminal-like	Reference	Reference	Reference	Reference
- HER2-positive	1.393 (0.893–2.173) 0.144	1.473 (0.935–2.319) 0.095	4.082 (2.228–7.476) 0.001 ^a^	4.746 (2.183–10.318) 0.001 ^a^
- Triple-negative				
Dimension post-NAT (mm)				
- ≤9	Reference	Reference	Reference	Reference
- >9	1.421 (0.772–2.613) 0.259	1.444 (0.782–2.666) 0.240	0.566 (0.285–1.124) 0.104	0.804 (0.348–1.856) 0.609
Stage post-NAT				
- pCR	Reference	Reference	Reference	n/a
- no pCR	0.282 (0.097–0.815) 0.019 ^a^	0.295 (0.102–0.855) 0.025 ^a^	0.041 (0.005–0.351) 0.004 ^a^	
- ypN0	Reference	Reference	Reference	Reference
- ypN+	1.281 (1.020–1.611) 0.034 ^a^	1.285 (1.022–1.617) 0.032 ^a^	1.493 (1.145–1.946) 0.003 ^a^	1.493 (1.086–2.053) 0.014 ^a^
Vascular invasion				
- Yes	Reference	Reference	Reference	Reference
- No	1.142 (0.643–2.029) 0.650	1.111 (0.625–1.997) 0.720	2.497 (1.317–4.734) 0.005 ^a^	1.990 (0.942–4.205) 0.071
Treatment				
Operation				
- BCS	Reference	Reference	Reference	Reference
- Mastectomy	1.070 (0.616–1.861) 0.809	1.080 (0.624–1.872) 0.783	1.020 (0.539–1.932) 0.951	1.541 (0.677–3.510) 0.303
- SLNB	Reference	Reference	Reference	Reference
- dALND	0.356 (0.190–0.667) 0.001 ^a^	0.376 (0.200–0.708) 0.002 ^a^	0.244 (0.107–0.556) 0.001 ^a^	0.228 (0.083–0.626) 0.004 ^a^
Adjuvant radiotherapy				
- Yes	Reference	Reference	Reference	Reference
- No	0.652 (0.350–1.212) 0.176	0.665 (0.357–1.239) 0.199	0.472 (0.233–0.956) 0.037 ^a^	0.468 (0.208–1.054) 0.067
Adjuvant chemotherapy				
- Yes	Reference	Reference	Reference	Reference
- No	1.357 (0.729–2.527) 0.336	1.381 (0.733–2.604) 0.318	0.970 (0.446–2.110) 0.938	0.954 (0.385–2.363) 0.920
Endocrine therapy				
- Yes	Reference	Reference	Reference	Reference
- No	0.630 (0.315–1.258) 0.190	0.685 (0.342–1.374) 0.287	1.801 (0.672–4.828) 0.242	1.833 (0.489–6.866) 0.368
T-DM1				
- Yes	Reference	Reference	Reference	Reference
- No	0.917 (0.546–1.538) 0.742	0.911 (0.540–1.539) 0.728	0.311 (0.155–0.624) 0.001 ^a^	0.209 (0.084–0.521) 0.001 ^a^

Footnotes: RFS: Recurrence-free survival, DDFS: Distant disease-free survival, OS: Overall survival, BCSS: Breast cancer-specific survival, HR: Hazard ratio, 95% CI: 95% Confidence interval, NAT: Neoadjuvant therapy, HER2: HER2 evaluated either on immunohistochemistry or on in situ hybridization, according to the ASCO CAP guidelines, pCR: Pathologic complete response, BCS: Breast-conserving surgery, SLNB: Sentinel lymph node biopsy, dALND: Direct axillary lymph node dissection, T-DM1: Trastuzumab–emtansine, n/a: not applicable, ^a^: Statistically significant.

## Data Availability

Data supporting reported results can be found in Appendix A.

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
