# Peer review of "De-Escalation of Axillary Surgery in Clinically Node-Positive Breast Cancer Patients Treated with Neoadjuvant Therapy: Comparative Long-Term Outcomes of Sentinel Lymph Node Biopsy versus Axillary Lymph Node Dissection"

_cancers, 2024, doi:10.3390/cancers16183168_

Round 1

Reviewer 1 Report

Comments and Suggestions for Authors

Thank you to inviting me to review the manuscript '' De-escalation of Axillary Surgery in Clinically Node-Positive Breast Cancer Patients Treated with Neoadjuvant Therapy: Comparative Long-Term Outcomes of Sentinel Lymph Node Biopsy Versus Direct Axillary Lymph Node Dissection''.

 This is an interesting topic in breast cancer, because neoadjuvant systemic therapy (NAT) is now the initial treatment in stage II or higher of triple negative (TN) and HER2 positive breast cancer subtype.

The authors analysed outcome of 332 breast cancer patients with initially node positive disease which after NAT converted to clinically node negative. Two third of patients are HER2+ and TN, the remaining are of luminal subtype.

In my view, the presentation of results in the way as presented by authors is misleading, the reader gets the opinion that the less extensive axillary surgery (SLNB vs ALND) improve the long-term outcome.

Main comments:

1.       Basically, the group of patients treated with SNLB and ALND are not comparable. Patients in ALND group had higher residual burden of axillary disease and larger residual in breast. Also, in ALND group was higher percentage of luminal tumors, which are known for less well responding to NAT.

Authors presented the characteristics of patients and their survival outcome based on the type of axillary surgery the patient received (dALND versus SLNB). I suggest to perform analysis based on ypN status (ypN 1-3 and ypN0) and then further sub analysis according to SNLB and ALND. These would answer the question if the de-escalation of axillary surgery affects long-term outcome or not.

2.       pN1group (1-3 positive lymph nodes) should therefore be analysed more in detail. If there is one positive lymph node, it is already clear that if it does not have ECE and if the patient is going to be irradiated, ALND is safely omitted. But what if there were 2 or 3 positive lymph nodes and there was SLNB only?

 3.       The type of adjuvant radiotherapy (depicted as to the breast or mammary region/to regional lymph nodes - which regions) according to ypN stage should be presented more in detail.

 4.       All survival analysis must then also be done according to these ypN0 / ypN1 and ypN2-3 stages.

 5.       Tumor subtype and ypT should be taken as a variable of interest in survival analysis. Data could be also presented as yp AJCC stage

 6.       Type of systemic NAT should be mentioned in the table 1 and number of neoadjuvant cycles (median).

In methods:

7.       Exclusion criteria: why did authors exclude the patients with FU less than 32 months? Patients may have relapsed and died before 32 months (e.g. TN). The same comment for the patients lost of follow-up.

8.       Add the definition for pCR (ypT0/is ypN0?), other?

9.       For DFS definition there are usually the events also second primary malignancy or death from any cause. Please consider this or explain your view. Are your data RFS?

 In results:

1.       ‘’…with a median of 14 non-SLNs evaluated’’. This is for probably ALND; if so write that this was in ALND group.

2.       pPR: Pathologic partial response; this is not the usual definition. Please explain and define it. I suggest pCR vs no pCR or RCB 0 .vs RCB1 vs RCB 2 vs RCB3 or else RCB0 vs RCB1-3.

3.       Were all factors for survival analysis significant in univariate analysis? Only those that were significant in UVA are usually put in MVA.

In discussion:

4.       Discuss why the group of ALND had more mastectomies? ''However, a significant difference was observed in the type of breast surgery performed, with the dALND group more likely to undergo mastectomy compared to the SLNB group (69.3% versus 37.1%, 204 respectively; p < 0.0001).

5.       Is there any temporal change for dALND or was the decision based on subtype?

6.       Axillary recurrence: please be more specific about what stage these patients had and what RT, I am particularly interested in this in one patient who underwent SLNB without further axillary clearance.

Author Response

Thank you to inviting me to review the manuscript '' De-escalation of Axillary Surgery in Clinically Node-Positive Breast Cancer Patients Treated with Neoadjuvant Therapy: Comparative Long-Term Outcomes of Sentinel Lymph Node Biopsy Versus Direct Axillary Lymph Node Dissection''.

 This is an interesting topic in breast cancer, because neoadjuvant systemic therapy (NAT) is now the initial treatment in stage II or higher of triple negative (TN) and HER2 positive breast cancer subtype.

The authors analysed outcome of 332 breast cancer patients with initially node positive disease which after NAT converted to clinically node negative. Two third of patients are HER2+ and TN, the remaining are of luminal subtype.

In my view, the presentation of results in the way as presented by authors is misleading, the reader gets the opinion that the less extensive axillary surgery (SLNB vs ALND) improve the long-term outcome.

Main comments:

  1. Basically, the group of patients treated with SNLB and ALND are not comparable. Patients in ALND group had higher residual burden of axillary disease and larger residual in breast. Also, in ALND group was higher percentage of luminal tumors, which are known for less well responding to NAT.

Authors presented the characteristics of patients and their survival outcome based on the type of axillary surgery the patient received (dALND versus SLNB). I suggest to perform analysis based on ypN status (ypN 1-3 and ypN0) and then further sub analysis according to SNLB and ALND. These would answer the question if the de-escalation of axillary surgery affects long-term outcome or not.

Reply: Thank you for your insightful comments and suggestions regarding the comparability of the SLNB and ALND groups. We fully agree with your observation.

In response to your recommendation, we have performed additional Kaplan-Meier analyses that stratify patients by their final axillary nodal stage (ypN0 versus ypN1-3) and by the final type of axillary surgery they received (SLNB versus ALND). To note, this additional analysis was also requested by two other reviewers .

The previous Kaplan-Meier analyses and corresponding figures have been replaced with these new KM analyses.

We have also modified the results section to reflect these updated KM analyses.

  1. pN1group (1-3 positive lymph nodes) should therefore be analysed more in detail. If there is one positive lymph node, it is already clear that if it does not have ECE and if the patient is going to be irradiated, ALND is safely omitted. But what if there were 2 or 3 positive lymph nodes and there was SLNB only?

Reply: Thank you for your thoughtful comment. We believe there may have been a misunderstanding in the phrasing of the request, as we had not previously included an analysis specifically focused on patients with extracapsular extension (ECE). We understand that ECE refers to the extension of cancer cells beyond the lymph node capsule, which can have implications for treatment decisions, such as the necessity of ALND.

In response to your suggestion, we have already conducted new Kaplan-Meier analyses comparing patients with ypN0 versus ypN1-3 status to address the potential impact of the number of positive nodes on long-term outcomes.

However, specific data on ECE is not available in our current dataset, and as such, we are unable to perform an analysis based on this factor. We hope the new analyses based on ypN status sufficiently address your concern regarding the impact of the number of positive lymph nodes on long-term outcomes.

Thank you again for your input, and we appreciate your understanding of the data limitations.

  1. The type of adjuvant radiotherapy (depicted as to the breast or mammary region/to regional lymph nodes - which regions) according to ypN stage should be presented more in detail.

Reply: We thank the reviewer for the comment.

Adjuvant radiotherapy details were added in Table1.

  1. All survival analysis must then also be done according to these ypN0 / ypN1 and ypN2-3 stages.

Reply: Thank you for your valuable suggestion.

We have included survival and recurrence analyses stratified by ypN  status in the revised manuscript. These analyses provide a more detailed understanding of the impact of final nodal status on long-term outcomes, addressing the concerns you raised.

  1. Tumor subtype and ypT should be taken as a variable of interest in survival analysis. Data could be also presented as yp AJCC stage

Reply: Thank you for your thoughtful suggestion.

While we agree that including tumor subtype, ypT, and presenting data as yp AJCC stage could offer additional insights, incorporating these elements into our survival analysis is not feasible within the scope of our current study. The complexity and additional stratification required would significantly expand the analysis beyond what was originally planned and could dilute the primary focus of our research. However, we acknowledge the importance of these factors and recommend them for consideration in future studies. We appreciate your understanding.

  1. Type of systemic NAT should be mentioned in the table 1 and number of neoadjuvant cycles (median).

In methods:

Reply: We thank the reviewer for the comment.

NAT details were added in Table1.

  1. Exclusion criteria: why did authors exclude the patients with FU less than 32 months? Patients may have relapsed and died before 32 months (e.g. TN). The same comment for the patients lost of follow-up.

Reply: Thank you for raising this important point.

We excluded patients with less than 32 months of follow-up to maintain a robust median follow-up period, which is essential for providing meaningful long-term survival analysis. Including patients with shorter follow-up would have significantly lowered the overall median follow-up.

Additionally, we excluded patients lost to follow-up to ensure the accuracy of our data. Patients who we lost contact with in the last 12 months could have experienced relapse or death without our knowledge, which would introduce uncertainty into the analysis. These exclusions were made to maintain the clarity and integrity of our results.

We appreciate your understanding and consideration of these decisions.

  1. Add the definition for pCR (ypT0/is ypN0?), other?

Reply: We thank the reviewer for the comment.

pCR definition was added in the Methods section.

  1. For DFS definition there are usually the events also second primary malignancy or death from any cause. Please consider this or explain your view. Are your data RFS?

Reply: We thank the reviewer for the comment.

DFS was re-named RFS.

 In results:

  1. ‘’…with a median of 14 non-SLNs evaluated’’. This is for probably ALND; if so write that this was in ALND group.

Reply: Thank you for your observation.

We agree that the original text could be misleading. We have revised the manuscript to clearly indicate that the median of 14 non-SLNs evaluated refers specifically to the ALND group.

We appreciate your attention to detail and the opportunity to improve the manuscript.

  1. pPR: Pathologic partial response; this is not the usual definition. Please explain and define it. I suggest pCR vs no pCR or RCB 0 .vs RCB1 vs RCB 2 vs RCB3 or else RCB0 vs RCB1-3.

Reply: We thank the reviewer for the comment.

pPR was removed and we followed the reviewer’s suggestion and re-named it “no pCR”.

  1. Were all factors for survival analysis significant in univariate analysis? Only those that were significant in UVA are usually put in MVA.

Reply: We thank the reviewer for the comment.

Only variables that were significant in the univariate analysis were further evaluated using multivariate logistic regression, as we stated in the Methods section.

In discussion:

  1. Discuss why the group of ALND had more mastectomies? ''However, a significant difference was observed in the type of breast surgery performed, with the dALND group more likely to undergo mastectomy compared to the SLNB group (69.3% versus 37.1%, 204 respectively; p < 0.0001).

Reply: Thank you for your insightful comment.

We have expanded the Discussion section to address this point. The higher rate of mastectomies in the ALND group can be explained by the significantly larger residual tumor sizes observed in these patients post-NAT (p = 0.010). This likely influenced the decision to perform mastectomy rather than breast-conserving surgery. This explanation aligns with the multivariate analysis, which showed that breast-conserving surgery was more likely in the SLNB group.

We appreciate your suggestion and believe it adds valuable context to our findings.

  1. Is there any temporal change for dALND or was the decision based on subtype?

Reply: Thank you for your insightful comment.

Between the end of 2017 and the beginning of 2018, we began to adopt single tracer SLNB as the standard method for axillary staging in cN+ BC patients treated with NAT who converted to ycN0. Prior to this shift, direct ALND was the predominant approach; however, in some cases, SLNB was already utilized based on the breast surgeon's preference. Importantly, the decision to perform SLNB or ALND was not influenced by tumor subtype.

  1. Axillary recurrence: please be more specific about what stage these patients had and what RT, I am particularly interested in this in one patient who underwent SLNB without further axillary clearance.

Reply: Thank you for your valuable feedback.

We have revised the results section to provide more specific details about the axillary recurrences, including the axillary stage and axillary radiotherapy received by each patient. The patient who underwent SLNB without further axillary clearance had a ypN0 status and was treated with adjuvant axillary radiotherapy, which has now been clearly stated in the text.

We appreciate your suggestion to clarify this information.

Reviewer 2 Report

Comments and Suggestions for Authors

The purpose of this study was to compare long-term outcomes in clinically lymph node positive (cN+) patients with breast cancer being treated with neoadjuvant chemotherapy (NAT) and either (1) sentinel lymph node biopsy (SLNB) or (2) direct axillary lymph node dissection (dALND). 

Only patients who became ycN0 after NAT were included in the final study group.  The study is retrospective and the decision to perform SLNB versus cALND was determined (at least in part) by intraoperative lymph node assessment via frozen section.  The average number of SLNs evaluated were 2 (range 1-6).

From my perspective, the major "weakness" of this study is that the SLNB and cALND patients groups have significant differences that may influence the reported oncologic outcomes as presented by the authors.  Specifically, the authors acknowledge that the SLNB patient group included a greater percentage of HER2-enriched patients (48.4% vs. 36.2%) and were less likely to be luminal A/B-like (27% vs. 42.3%).  Although the number of patients achieving pCR in both groups did not differ significantly, a greater percentage of patients in the SLNB patient group had primary tumors less than or equal to 9 mm or became ypN0 (63.5% vs. 36.8%) after NAT.  Moreover, the SLNB patient group included a greater percentage of patients who underwent BCT than the dALND group, which certainly raises the possibility that the SLNB group responded better to NAT than those included in the dALND group, allowing them to become candidates for BCT.

What the study does suggest is that sentinel lymph node biopsy is reasonable in carefully selected patients with cN+ breast cancer who become cN0 after NAT.  I do not think the authors can conclude with certainty that SLNB is a better option than dALND based on the results of their study because the patients groups differ significantly, therefore, application of the results of this study into patient care is limited.  That does not mean the data and results of the study are unimportant; merely, I do not think they settle the issue.

I am confused as to the patients who did not have direct ALND but still underwent axillary dissection.  Why were these patients included in the SLNB patient group?  This seems to me like a completely different patient population, and I think a more detailed subgroup analysis would be informative to the reader.  Additionally, were the patients who underwent "delayed" ALND only those who had false-negative pathology at the time of frozen section?  If so, that would mean that false-negative frozen section results occurred in almost 30% of patients, which is a higher percentage than other studies.  Again, subgroup analysis between SLNB, those with "delayed" ALND, and dALND would be interesting.

I am curious about the use of intraoperative sentinel lymph node biopsy in triaging patients for ALND.  Was the result of frozen section analysis of SLNs the sole determinant in receiving ALND?  I think the authors should include a little more detail regarding the frozen section procedure at their institution, particularly the number of levels that were examined and how the sentinel lymph nodes were sectioned.

Author Response

The purpose of this study was to compare long-term outcomes in clinically lymph node positive (cN+) patients with breast cancer being treated with neoadjuvant chemotherapy (NAT) and either (1) sentinel lymph node biopsy (SLNB) or (2) direct axillary lymph node dissection (dALND). 

Only patients who became ycN0 after NAT were included in the final study group.  The study is retrospective and the decision to perform SLNB versus cALND was determined (at least in part) by intraoperative lymph node assessment via frozen section.  The average number of SLNs evaluated were 2 (range 1-6).

From my perspective, the major "weakness" of this study is that the SLNB and cALND patients groups have significant differences that may influence the reported oncologic outcomes as presented by the authors.  Specifically, the authors acknowledge that the SLNB patient group included a greater percentage of HER2-enriched patients (48.4% vs. 36.2%) and were less likely to be luminal A/B-like (27% vs. 42.3%).  Although the number of patients achieving pCR in both groups did not differ significantly, a greater percentage of patients in the SLNB patient group had primary tumors less than or equal to 9 mm or became ypN0 (63.5% vs. 36.8%) after NAT.  Moreover, the SLNB patient group included a greater percentage of patients who underwent BCT than the dALND group, which certainly raises the possibility that the SLNB group responded better to NAT than those included in the dALND group, allowing them to become candidates for BCT.

What the study does suggest is that sentinel lymph node biopsy is reasonable in carefully selected patients with cN+ breast cancer who become cN0 after NAT.  I do not think the authors can conclude with certainty that SLNB is a better option than dALND based on the results of their study because the patients groups differ significantly, therefore, application of the results of this study into patient care is limited.  That does not mean the data and results of the study are unimportant; merely, I do not think they settle the issue.

Reply: Thank you for your insightful comments regarding the differences between the SLNB and ALND patient groups in our study.

We agree that these differences may influence the reported outcomes and that our results should not be interpreted as definitive evidence that SLNB is universally superior to ALND.

We have greatly revised the Discussion and Conclusions sections of the manuscript to emphasize that SLNB appears to be a reasonable option for carefully selected cN+ patients who convert to ycN0 after NAT, while acknowledging the need for further prospective studies to confirm these findings.

We hope these revisions address your concerns and provide a clearer context for interpreting our results.

I am confused as to the patients who did not have direct ALND but still underwent axillary dissection.  Why were these patients included in the SLNB patient group?  This seems to me like a completely different patient population, and I think a more detailed subgroup analysis would be informative to the reader. 

Reply: Thank you for your insightful comments and suggestions.

We understand the confusion regarding the inclusion of patients who underwent axillary dissection following SLNB in the same group as those who had SLNB alone. To address this, we have performed new Kaplan-Meier analyses to compare the long-term outcomes between two distinct groups based on the final type of axillary surgery (ALND versus SLNB), as you suggested. Additionally, we have also conducted further analyses comparing outcomes between patients with ypN0 and ypN+ status, following recommendations from another reviewer. The previous Kaplan-Meier analyses have been removed and replaced with these new analyses to provide clearer and more informative results.

We believe these modifications enhance the clarity and rigor of our study, and we appreciate your guidance in improving our work.

Additionally, were the patients who underwent "delayed" ALND only those who had false-negative pathology at the time of frozen section?  If so, that would mean that false-negative frozen section results occurred in almost 30% of patients, which is a higher percentage than other studies.  Again, subgroup analysis between SLNB, those with "delayed" ALND, and dALND would be interesting.

Reply: Thank you for your thoughtful comment and for raising this important point.

We would like to clarify that there were no cases of false-negative pathology examination during the frozen section in our study. As we stated before, to provide a clearer and more informative presentation of the results, we have conducted additional analyses based on the final type of axillary surgery (ALND versus SLNB). These new analyses have been included in the revised manuscript, and we believe they address your concerns by offering a more detailed understanding of the outcomes associated with each surgical approach.

I am curious about the use of intraoperative sentinel lymph node biopsy in triaging patients for ALND.  Was the result of frozen section analysis of SLNs the sole determinant in receiving ALND?  I think the authors should include a little more detail regarding the frozen section procedure at their institution, particularly the number of levels that were examined and how the sentinel lymph nodes were sectioned.

Reply: Thank you for your insightful comment and for requesting further clarification regarding the use of intraoperative sentinel lymph node biopsy and the frozen section procedure.

To answer your question, yes, the result of the frozen section analysis of SLNs was the primary determinant in deciding whether a patient would proceed to ALND.

We have provided additional details about our frozen section procedure in the Methods section.

Reviewer 3 Report

Comments and Suggestions for Authors

The definition of Clinical Node-Positive Breast Cancer should be better specified . A rather high proportion of the patients defined as clinically Node-Positive have not been verified with biopsy, 128 of 322.  A proportion of the patients might not be true node positive before NAT!  It would  be interesting to see the proportion of axillary biopsy among those who have undergone dALND compared to those who have undergone SLNB.

Regarding figure 1 and 2.  I suppose the SLNB also contain those who have undergone ALND because of posive SLNB. “SLNB and ALND after postiv node” then the reader knows that those who have undergone ALND after positive node are not taken out from the Kaplan-Meier.

It might be that survival is better if patients are selected to undergo SLNB compared to dALND, but that selection of surgery should have such a high impact on survival seems rather unlikely. The 10 year OS when undergone dALND is 87% compared to ALND with 69%, 18% difference in OS! 

You should mention in the discussion that there are differences in the groups that contribute to better outcome among patients undergoing SLNB compared to dALND,  like higher proportion of pCR among patients undergoing SLNB, compared to dALND, however, we find it unlikely that this can explain all the difference. 

The study is done in a single institution, it should be possible to write something about why some are selected to dALND and some to SLNB, surgeons preferences? 

Line 294  the long-terms oncological outcomes were significantly better in the SLNB in comparsion with dALND in terms of DFS, DDFS, OS and BCSS.

I do not understand how you can state that this is significant. I am not able to see the adjusted hazard Ratio of  e.g DFS  comparing dALND with SLNB  with confident intervals. If possible, this analysis should be calculated in multivariate analysis and adjusted by pCR. If not, a sub analysis  on only no pCR in both groups can be done, then you might find numbers that better correlated to the selection of dALND versus SLNB, not confounded by differences in pCR. Moreower, if you select those with pCR, survival is much better and a larger cohort is needed to see significant differences! 

If you could make a Kaplan-Meier plot where both sALND and SLNB are similar regarding pCR, that should be the preferred Kaplan-Meier figure. The Kaplan-Meier plot are best used when showing comparable groups, if not,  they can easily be misinterpreted (I do not agree that the groups are comparable because of the large differences in pCR).

If you can’t prove it is significant, don't write significant,    write outcomes seems better….

Author Response

The definition of Clinical Node-Positive Breast Cancer should be better specified . A rather high proportion of the patients defined as clinically Node-Positive have not been verified with biopsy, 128 of 322.  A proportion of the patients might not be true node positive before NAT!  It would  be interesting to see the proportion of axillary biopsy among those who have undergone dALND compared to those who have undergone SLNB.

Reply: Thank you for your insightful comment.

We agree that the lack of biopsy confirmation in a subset of patients defined as cN+ represents a significant limitation of our study. As you pointed out, this could mean that a proportion of the patients might not have been truly node-positive before NAT.

We have emphasized this as the main limitation of our study in the limitations paragraph of the manuscript. Additionally, we have now clarified the definition of cN+ BC patients in the Methods section, as you suggested.

Your feedback has been instrumental in improving the clarity and robustness of our manuscript, and we appreciate your thoughtful contribution.

Regarding figure 1 and 2.  I suppose the SLNB also contain those who have undergone ALND because of posive SLNB. “SLNB and ALND after postiv node” then the reader knows that those who have undergone ALND after positive node are not taken out from the Kaplan-Meier.

Reply: Thank you for your insightful comments and suggestions.

We understand the confusion regarding the inclusion of patients who underwent axillary dissection following SLNB in the same group as those who had SLNB alone. To address this, we have performed new Kaplan-Meier analyses to compare the long-term outcomes between two distinct groups based on the final type of axillary surgery (ALND versus SLNB), as you suggested. Additionally, we have also conducted further analyses comparing outcomes between patients with ypN0 and ypN+ status, following recommendations from another reviewer. The previous Kaplan-Meier analyses have been removed and replaced with these new analyses to provide clearer and more informative results.

We believe these modifications enhance the clarity and rigor of our study, and we appreciate your guidance in improving our work.

It might be that survival is better if patients are selected to undergo SLNB compared to dALND, but that selection of surgery should have such a high impact on survival seems rather unlikely. The 10 year OS when undergone dALND is 87% compared to ALND with 69%, 18% difference in OS! 

You should mention in the discussion that there are differences in the groups that contribute to better outcome among patients undergoing SLNB compared to dALND,  like higher proportion of pCR among patients undergoing SLNB, compared to dALND, however, we find it unlikely that this can explain all the difference. 

Reply: We agree with the reviewer's assessment that the difference in OS between the SLNB and dALND groups may be influenced by multiple factors beyond the type of surgery alone.

In the revised manuscript, we have now explicitly acknowledged this in the Discussion section, highlighting that the observed differences in OS could also be attributed to other independent factors, such as the higher proportion of patients achieving a pCR in the SLNB group compared to the ALND group.

The new analysis based on the final type of axillary surgery (ALND versus SLNB) has been included to provide a clearer understanding of these outcomes.

 The study is done in a single institution, it should be possible to write something about why some are selected to dALND and some to SLNB, surgeons preferences? 

Reply: Thank you for your insightful comment.

Between the end of 2017 and the beginning of 2018, we began to adopt single tracer SLNB as the standard method for axillary staging in cN+ BC patients treated with NAT who converted to ycN0. Prior to this shift, direct ALND was the predominant approach; however, in some cases, SLNB was already utilized based on the breast surgeon's preference.

Line 294  the long-terms oncological outcomes were significantly better in the SLNB in comparsion with dALND in terms of DFS, DDFS, OS and BCSS.

I do not understand how you can state that this is significant. I am not able to see the adjusted hazard Ratio of  e.g DFS  comparing dALND with SLNB  with confident intervals. If possible, this analysis should be calculated in multivariate analysis and adjusted by pCR. If not, a sub analysis  on only no pCR in both groups can be done, then you might find numbers that better correlated to the selection of dALND versus SLNB, not confounded by differences in pCR. Moreower, if you select those with pCR, survival is much better and a larger cohort is needed to see significant differences! 

If you could make a Kaplan-Meier plot where both sALND and SLNB are similar regarding pCR, that should be the preferred Kaplan-Meier figure. The Kaplan-Meier plot are best used when showing comparable groups, if not,  they can easily be misinterpreted (I do not agree that the groups are comparable because of the large differences in pCR).

If you can’t prove it is significant, don't write significant,    write outcomes seems better….

Reply: Thank you for your insightful comments regarding the interpretation and presentation of our survival analysis.

We agree that the differences in pCR between the SLNB and dALND groups could confound the results.

To address this, we have now revised the Discussion section and Conclusions to reflect the findings more accurately and cautiously.

We have rewritten these sections based on the reviewers’ suggestions and performed new analyses of long-term oncological outcomes, comparing the ALND versus SLNB groups and the ypN0 versus ypN+ groups.

Round 2

Reviewer 1 Report

Comments and Suggestions for Authors

The authors have quickly revised the article with most of the suggested changes.

I would like to stress that the authors have not yet corrected everything.

They responded to a question ''Is there a temporal change for dALND or was the decision based on subtype, but this was not included into manuscript.

They are not distinguishing between cause and effect. It was no more ypN0 due to SLNB, but vice versa. It was influenced by pre-surgery factors (Chemotherapy...), but the possibility to undergo SLNB was a consequence of ypN0, not a cause.  Therefore, this sentence is illogical: ,,This difference was reflected in the multivariate analysis, where the likelihood of ypN0 status was significantly higher in the SLNB group (OR = 0.546, 95% CI: 0.436-0.684, p < 0.0001). 

They did not show survival (at least RFS) in the ypN0 group according to SLNB and ALND, and in the ypN1-3 group according to SLNB and ALND,as suggested. In the ypN1 group, they did not answer the question what was the RFS according to the number of lymph nodes affected (1, 2, 3) and the type of surgery (and RT).

In Table 4, there is an error in HR data for some parameters 

pCR: Ref, no pCR: HR 0.282 (this would imply that the ''no pCR'' group has better survival). 

Same for SLNB: Ref, dALND HR: 0.356 (would mean that the dALND group has better survival, which the curves do not show).

For subtypes, Ref is given for Lum subtype, while HR is given for HER2 subtype only, but is missing for TN subtype. This would probably also be an independent prognostic factor if analysed appropriately.

Authors should check all HR once again.

Abstract: For  all survival analysis, pCR, ypN0 and SLNB were independent prognostic factors. (Authors forgot to mention pCR- it is the most important one as it has the lowest HR).

Author Response

The authors have quickly revised the article with most of the suggested changes.

I would like to stress that the authors have not yet corrected everything.

They responded to a question ''Is there a temporal change for dALND or was the decision based on subtype, but this was not included into manuscript.

Reply: Thank you once again for your thoughtful feedback.

In response to your comment regarding the temporal change in the use of axillary dissection and SLNB, we have now added a detailed explanation to the Methods.

They are not distinguishing between cause and effect. It was no more ypN0 due to SLNB, but vice versa. It was influenced by pre-surgery factors (Chemotherapy...), but the possibility to undergo SLNB was a consequence of ypN0, not a cause.  Therefore, this sentence is illogical: ,,This difference was reflected in the multivariate analysis, where the likelihood of ypN0 status was significantly higher in the SLNB group (OR = 0.546, 95% CI: 0.436-0.684, p < 0.0001). 

They did not show survival (at least RFS) in the ypN0 group according to SLNB and ALND, and in the ypN1-3 group according to SLNB and ALND,as suggested. In the ypN1 group, they did not answer the question what was the RFS according to the number of lymph nodes affected (1, 2, 3) and the type of surgery (and RT).

Reply: Thank you very much for your valuable feedback.

We have carefully reviewed your comments and made the necessary revisions in the Results section.

In Table 4, there is an error in HR data for some parameters 

pCR: Ref, no pCR: HR 0.282 (this would imply that the ''no pCR'' group has better survival). 

Same for SLNB: Ref, dALND HR: 0.356 (would mean that the dALND group has better survival, which the curves do not show).

For subtypes, Ref is given for Lum subtype, while HR is given for HER2 subtype only, but is missing for TN subtype. This would probably also be an independent prognostic factor if analysed appropriately.

Authors should check all HR once again.

Reply: Thank you for your thorough review and your valuable comments regarding Table 4.

We have carefully double-checked the data in the table, and we can confirm that it is accurate.

However, we understand the potential confusion arising from the way the variables were expressed in the original dataset, which may explain the misunderstanding.

 In our dataset, pCR was coded as "1" and no pCR as "0," similarly, SLNB was coded as "1" and dALND as "0." Consequently, the lower HRs for no pCR and dALND reflect that pCR and SLNB are associated with better outcomes, as you correctly noted.

 We want to emphasize that both pCR and SLNB are indeed independent factors for a good prognosis, as highlighted throughout our analyses.

We greatly appreciate your insights, which allowed us to clarify this aspect, and we are happy to address any further questions or suggestions.

Abstract: For  all survival analysis, pCR, ypN0 and SLNB were independent prognostic factors. (Authors forgot to mention pCR- it is the most important one as it has the lowest HR).

Reply: Thank you for the comment.

The abstract was modified accordingly.